# The PiNe box: Development and validation of an electronic device to time-lock multimodal responses to sensory stimuli in hospitalised infants

Alan Worley[1], Kirubin Pillay[2], Maria M. Cobo[2], Gabriela Schmidt Mellado[2], Marianne van der Vaart[2], Aomesh Bhatt[2], Caroline Hartley[2]*

1 Department of Clinical Neurophysiology, Great Ormond Street Hospital for Children, London, United Kingdom, 2 Department of Paediatrics, University of Oxford, Oxford, United Kingdom

* caroline.hartley@paediatrics.ox.ac.uk

## Abstract

Recording multimodal responses to sensory stimuli in infants provides an integrative approach to investigate the developing nervous system. Accurate time-locking across modalities is essential to ensure that responses are interpreted correctly, and could also improve clinical care, for example, by facilitating automatic and objective multimodal pain assessment. Here we develop and assess a system to time-lock stimuli (including clinically-required heel lances and experimental visual, auditory and tactile stimuli) to electrophysiological research recordings and data recorded directly from a hospitalised infant's vital signs monitor. The electronic device presented here (that we have called 'the PiNe box') integrates a previously developed system to time-lock stimuli to electrophysiological recordings and can simultaneously time-lock the stimuli to recordings from hospital vital signs monitors with an average precision of 105 ms (standard deviation: 19 ms), which is sufficient for the analysis of changes in vital signs. Our method permits reliable and precise synchronisation of data recordings from equipment with legacy ports such as TTL (transistor-transistor logic) and RS-232, and patient-connected networkable devices, is easy to implement, flexible and inexpensive. Unlike current all-in-one systems, it enables existing hospital equipment to be easily used and could be used for patients of any age. We demonstrate the utility of the system in infants using visual and noxious (clinically-required heel lance) stimuli as representative examples.

## Introduction

Premature-born infants experience a plethora of sensory stimulation when exposed to the extrauterine environment. These may have negative consequences for the infant, including possible long-term neurodevelopmental effects [1–4]. In contrast, positive touch, such as breastfeeding or skin-to-skin contact, may be important for neurodevelopment [5] and may mitigate alterations in sensory processing related to noxious stimuli [6]. Visual, auditory,

**Data Availability Statement:** Data for Study 1 is provided as S1 Data file. Due to ethical constraints and the sensitive nature of medical data collection,

data for Study 2 and 3 are available from the corresponding author on request. Requests can be made directly to caroline.hartley@paediatrics.ox.ac. uk or directed to the Paediatric Neuroimaging Group, University of Oxford through the institutional email neonatal.research@ouh.nhs.uk.

**Funding:** This work was funded by the Wellcome Trust and Royal Society through a Sir Henry Dale Fellowship awarded to CH (Grant reference: 213486/Z/18/Z). The funders had no role in study design, data collection and analysis, decision to publish, or preparation of the manuscript.

**Competing interests:** KP, MMC, AB and CH are inventors/contributors to Patent Application Number: PCT/GB2022/052836: Systems and methods for measuring a response of a subject to an event. This patent is not directly related to the material presented in this paper, but relates to methods to analyse EEG responses to noxious stimuli. This does not alter our adherence to PLOS ONE policies on sharing data and materials. The authors declare no other competing interests.

tactile and noxious stimuli can all elicit responses at many levels of the nervous system, including increases in heart rate, whole body reflexes, and evoked cortical responses [7–12]. Investigating how an infant responds to a stimulus, at all levels of the nervous system, is essential to understand nervous system development and long-term effects.

Vital signs, such as heart rate, respiratory rate and oxygen saturation, are monitored routinely in hospitalised infants. Time-locking recordings of routine hospital monitoring to experimental or clinical stimuli (such as noxious, visual, tactile, auditory), and linking to research-related recordings of electrophysiological activity such as electroencephalography (EEG) and electromyography (EMG), provides a valuable approach to investigate multimodal sensory responses. Current methods to integrate EEG devices, vital signs devices and stimulus triggers exist but require purchase of all-in-one systems that generally offer a limited number of physiological recording channels, which could reduce research potential. Furthermore, integrating specific stimuli with such systems usually requires additional pre-processing devices to be added upstream to correctly pre-condition the triggers. Ultimately, this results in a solution that is costly and still requires multiple devices to be integrated together. Instead, a more ideal solution would be to develop a device that allows seamless integration of a broad range of available electrophysiological recording equipment and hospital vital sign monitors, with stimulus triggers utilising common network communication protocols in a cost-effective, efficient and robust manner. Furthermore, this approach would not be limited to electrophysiological and vital signs equipment but could be adopted by a range of standard clinical equipment, leading to greater resource utilisation for research studies.

Here, we describe the development of a new device known as the PiNe (Raspberry *Pi* to *Ne*twork) box, which integrates an existing event-detection interface, originally created to time-lock stimulus events to an electrophysiological research recording system [13]. The PiNe box additionally allows simultaneous time-locking of these events to recordings of data continuously downloaded from a hospital vital signs monitor. The PiNe box can connect equipment using legacy ports such as TTL (transistor-transistor logic) with modern network devices, communicating using established network protocols. It runs using a small low-cost and widely available (approximately $40) single-board computer (Raspberry Pi, Raspberry Pi Foundation). Moreover, it is flexible, with the ability to integrate different sensory stimulation equipment, research devices and clinical devices. We used three approaches to test the accuracy and clinical utility of the PiNe box. Firstly, we calculated the precision in time-locking of events on the vital signs recordings in a controlled laboratory setting. Secondly, we assessed the synchronisation of the recordings between the electrophysiological recording system and the vital signs monitor in a clinical setting. Finally, we demonstrated the utility of the system across multimodal recordings in infants.

## Materials and methods

### The PiNe box and event detection interface

Fig 1A shows a schematic of the overall system including the PiNe box. Overall, the system allows events (sensory stimuli or a push button device) to be simultaneously triggered and marked onto electrophysiological research recordings and recordings from a hospital vital signs monitor. An event detection interface for clinically-required heel lances and experimental tactile stimuli has been previously described [13] and can be used to generate these external triggers. Commercially available stimulators (such as auditory and visual stimulators) can also be easily integrated.

Our new device—the PiNe box—reads in TTL (transistor-transistor logic) triggers from the external triggering devices which are sent to its internal Raspberry Pi. These TTL triggers are

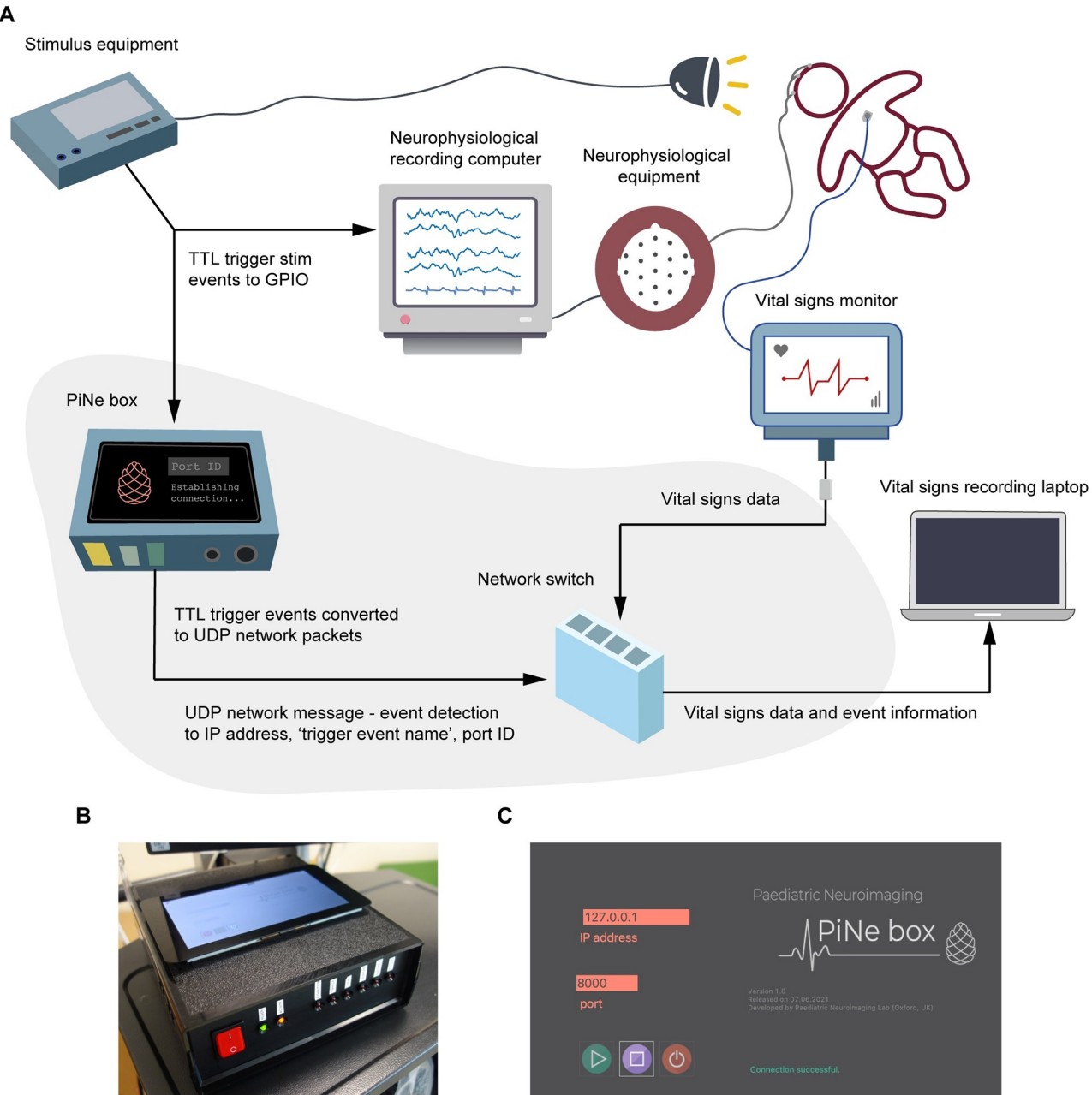

**Fig 1. The PiNe box.** (A) Schematic of the full system. Grey shading illustrates the newly developed parts of the monitoring system including the PiNe box and its associated connections. Abbreviations: TTL: Transistor-transistor logic, GPIO: General-Purpose Input Output, UDP: User Datagram Protocol, IP: Internet protocol. (B) A photograph of the PiNe box. (C) The graphical user interface (GUI) on the PiNe box.

simultaneously sent to the PiNe box and to the General-Purpose Input Output (GPIO) ports of the electrophysiological recording equipment (which allows simultaneous time-locking of stimulus onset to the electrophysiological activity, as described in previous publications [13]). The Raspberry Pi in the PiNe box converts the TTL signal to a UDP (User Datagram Protocol) message which is sent to an acquisition laptop that also continuously records the signals from the hospital vital signs monitor. The laptop is connected via an ethernet connection to the

PiNe box and to the vital signs monitor, simultaneously receiving the vital signs data and UDP messages. A network switch is used to allow these multiple ethernet connections to the laptop. A network isolator is connected between the hospital vital signs monitor and the network switch to ensure a secure connection to the hospital monitor.

To record the vital signs data from the hospital monitor the separate acquisition laptop uses ixTrend software (ixitos GmbH, Germany), but alternative software to record vital signs is available [14]. The ixTrend software has inbuilt functionality to mark external network events onto data recordings and we use this feature here to allow the transmitted UDP messages (i.e., the stimulus triggers) to be time-locked to the vital signs data. UDP network messages were used as the ixTrend software requires this protocol, however, TCP (Transmission Control Protocol) could also be straightforwardly used and is implemented in the PiNe box as an alternative option. The PiNe box includes a touch screen (run by the Raspberry Pi) which presents a Graphical User Interface (GUI) allowing the user to select the appropriate IP address and listening ports to link to the monitoring laptop and to determine whether there is a successful connection (Fig 1B and 1C).

A unique user-defined message can be generated for different triggers from the stimulus equipment so that different stimulus types can have independent event markers. We developed our system to receive and send six independent markers (clinically-required heel lance, experimental visual, tactile, auditory, and noxious stimuli, and a push button device which can be used to simultaneously manually mark events on both devices e.g. background periods or other clinical procedures).

The system has been approved for use within our hospital by the local clinical engineering team.

Running the GUI, establishing connections, reading of the input stimulus triggers, generation of the UDP output messages and transmission to the monitoring laptop are all controlled by a bespoke module that is written fully in Python 3. The overall process of the code is summarised in the pseudo code given in S1 Fig.

## Assessment of the PiNe box

**Study design and participants.** We assessed the use of the PiNe box in three studies. In Study 1 the precision of time-locking of events to the vital signs monitor was assessed in seven recordings of background noise (the monitor was not connected to an infant during these recordings). The push button was pressed onto one of the ECG electrodes to generate an artefact and a time-locked event mark. The time difference between the event mark and the artefact onset was used to calculate the precision in time-locking of events on the vital signs recordings.

In Study 2, the synchronisation of the event markers between the electrophysiological recordings and the recordings of the hospital vital signs monitor was investigated in recordings from 48 infants. Infant demographics are given in S1 Table.

In Study 3, recordings from two infants are presented as representative examples to demonstrate the utility of the system; visual-evoked responses are shown from an infant who was born at 32 weeks (+ 2 days) gestation with a birth weight of 1.66 kg and studied at 34 weeks (+ 4) postmenstrual age (PMA); noxious-evoked responses (recorded to a clinically-required heel lance) are shown from an infant who was born at 37 weeks' (+ 0) gestation with a birth weight of 2.14 kg and studied when they were 3 days old.

All infants were recruited as part of other ongoing studies. The infants were recruited from the Newborn Care Unit and Maternity wards of the John Radcliffe Hospital (Oxford University Hospitals NHS Foundation Trust, Oxford, United Kingdom). Ethical approval was

obtained from the National Research Ethics Service (references: 19/LO/1085, 12/SC/0447, 21/LO/0523) and parental written informed consent was obtained before each participant was studied. The studies were conducted in accordance with the standards set by the Declaration of Helsinki and Good Clinical Practice guidelines.

**Recording techniques.** Electrophysiological research recordings were acquired from DC to 800 Hz using a SynAmps RT 64-channel headbox and amplifiers (Compumedics Neuroscan). Activity was recorded using CURRY scan7 neuroimaging suite (Compumedics Neuroscan) with a sampling rate of 2000 Hz. EEG was recorded from eight electrode sites (Cz, CPz, C3, C4, Oz, FCz, T3, T4) according to the modified international 10–20 system with reference at Fz and ground at Fpz. The scalp was cleaned with preparation gel (Nuprep gel, D.O. Weaver and Co.) and disposable Ag/AgCl cup electrodes (Ambu Neuroline) were placed with conductive paste (Elefix EEG paste, Nihon Kohden). Electrocardiograph (ECG) was recorded on this system using a single electrode placed on the infant's chest, referenced to Fz and with the ground electrode at Fpz.

Infants' vital signs were monitored using a Philips IntelliVue MX800 or MX750 clinical monitor, and data was continuously downloaded from the monitor to an external laptop using ixTrend software (ixitos GmbH, Germany). Heart rate, oxygen saturation and respiratory rate (calculated by the monitor) were downloaded onto the laptop at a sampling rate of 1 Hz; the ECG was downloaded onto the laptop at a sampling rate of 250 Hz and recorded with three electrodes placed on the infant's chest, the impedance pneumograph was downloaded onto the laptop at a sampling rate of 62.5 Hz and recorded from the chest electrodes, and the photoplethysmography was downloaded onto the laptop at a sampling rate of 125 Hz from a probe placed on the infant's foot. The sampling rates of the signals are set by the Philips monitor rather than the software on the laptop; the PiNe box does not rely on certain sampling rates and could be used with clinical monitors with different sampling rates. Alternative software is available to download data from Philips or other monitors [14].

To assess the precision of time-locking to the recordings of the vital signs (Study 1), recordings of the impedance pneumograph were taken using three ECG electrodes attached to each other using conductive paste (Elefix EEG paste, Nihon Kohden) applied to a table—the electrodes were not connected to an infant during these recordings. This set-up allowed the push button device to be pressed on to the electrodes generating both an event mark and an artefact on the recordings, the timings of which could be compared. As the vital signs monitor has built-in filters that cannot be turned off, the electrodes needed to be connected to each other via conductive paste so that background noise and the push button artefact could be recorded. To assess replicability of the precision, we pressed the push button onto the electrodes 5 times per recording in 7 separate recordings.

**Stimuli.** Visual, tactile and clinically-required heel lances were performed in Studies 2 and 3. Infants were recruited as part of other ongoing research studies and so the type of stimulus presented was dependent on the particular research question. In study 2, on 15 test occasions visual stimuli were performed, on 7 occasions touch stimuli were performed and on 35 occasions clinically-required heel lances were performed.

*Visual stimulus*. Trains of approximately 10 visual stimuli were presented with an inter-stimulus interval of at least 10 seconds. This interval was extended if the infant was unsettled. The stimulus was a flash of light presented using a Lifelines Photic Stimulator (intensity level 4, approximately 514 lumens), which was held approximately 30 cm directly in front of the infant's eyes.

*Tactile stimulus*. Trains of approximately 10 tactile stimuli were presented with an inter-stimulus interval of at least 10 seconds. This interval was extended if the infant was unsettled.

The infant's foot was lightly tapped with a tendon hammer. The tendon hammer was modified to include a built-in force transducer (Brüel & Kjær) that sends a trigger pulse at the point of stimulation [13].

*Clinically-required heel lance*. Heel lances were only performed when required clinically to obtain a blood sample. Heel lances were performed on the medial or lateral plantar surface of the heel. Release of the lancet blade was time-locked to the recordings using the event detection interface previously described [13]. The single infant presented in Study 3 clinically-required a heel lance blood test to assess bilirubin levels (SBR). The lance was performed on the right foot (side chosen based on clinical judgement) using a Smiths Medical Neoheel Safety Lancet (Newborn, 1.0 mm Depth, model 1052N).

**Analysis.** *Study 1*: *Precision of time-locking of events to vital signs monitor*. Analysis was performed using MATLAB (R2021a, MathWorks). To assess the precision of time-locking of events to the vital signs monitor the push button was pressed onto one of the ECG electrodes simultaneously generating an event mark on the recording from the vital signs monitor and a visible artefact (S2 Fig). This process was repeated 5 times per recording, and in 7 recordings to assess replicability. As the push button was pressed on the ECG electrodes this generated an artefact on the ECG trace and the impedance pneumograph (which is a recording of changes in the resistance between ECG electrodes). The artefact was most visible on the impedance pneumograph and so this signal was used in the analysis (S2 and S3 Figs). Precision was calculated as the time difference between the event mark (generated using the PiNe box) and the start of the artefact. The signal was first epoched from 1 second before to 1 second after the event mark, and baseline corrected to the pre-event mark mean. The start of the artefact was calculated automatically as the point at which the signal went below a threshold defined as 3% of the minimum signal value in the 1 second after the event mark (S2 Fig). This threshold was chosen based on visual inspection of the data; in 2 files the start time was manually adjusted to better fit the data. Code for this analysis is provided as S1 Code and the associated data in S1 Data.

*Study 2*: *Precision of time-synchronisation between devices*. To assess the synchronisation of event markers across the two recording devices, the ECG signal was recorded on both devices and the cross-correlation calculated to identify the time differences between the signals. This was assessed in relation to event marks from visual, tactile and clinically-required noxious stimuli. The ECG signal from the Philips monitor (the clinical vital signs monitor) is filtered during acquisition using the settings built into the monitor itself (the filter was set to 'monitor' mode). To eliminate slow frequency drift due to movement artefacts and to make the filters comparable to those used for the ECG signal acquired using Curry 7 we additionally filtered the signal after acquisition from 12–40 Hz using EEGlab [15]. The ECG signal acquired using Curry 7 (Compumedics Neuroscan—the electrophysiological research recording device) was filtered after acquisition from 12–40 Hz using EEGlab, with a notch filter at 50 Hz, and down-sampled using the MATLAB *resample* function to a sampling rate of 250 Hz to match that of the signal from the Philips monitor. The ECG signal from both recordings were epoched 5 seconds before and after each stimulus event, according to the time of the event marked on the individual recordings. The alignment of the signals between the two devices (and consequently the alignment of the event time-locking) was calculated as the time corresponding to the maximum cross-correlation, and the values were visually checked for spurious results due to artefact and for good alignment between the shifted signals. For recording sessions where multiple stimuli were performed in a single recording session, the average time-difference between devices across all stimuli was calculated. A total of 51 test occasions were included in the analysis (recorded in 48 infants).

*Study 3*: *Demonstration of system utility*. To demonstrate the utility of the system in the simultaneous assessment of electrophysiological and vital signs responses we recorded EEG and vital signs responses to visual stimuli and a clinical-required heel lance, detailing the response in two separate infants. EEG signals were filtered from 0.5–30 Hz, with a notch filter at 50 Hz, using EEGlab [15]. The signal was epoched from 0.5 seconds before the stimulus until 1 second afterwards and baseline corrected to the pre-stimulus mean. In response to the heel lance, which was a single event, the EEG response is shown at the Cz electrode, and Woody filtered by ±100 ms to the previously described and validated template of noxious-evoked brain activity [16]. In response to the visual stimulus, the average response across 10 stimuli and following a single stimulus is shown at the Oz electrode, the electrode with the dominant response to the visual stimulus [17].

Heart rate and oxygen saturation responses were calculated using the values obtained directly from the vital signs monitors. A one-minute epoch (with 15 seconds before the stimulus) was taken for the single heel lance; for visual stimuli the data was epoched with 5 seconds before the stimulus to 10 seconds afterwards due to the inter-stimulus interval. To calculate the average response to the visual stimulus, values were first baseline corrected by subtracting the pre-stimulus mean.

Inter-breath intervals (IBIs) were calculated from the impedance pneumography (IP) using a previously described algorithm which is validated for use in infants and more sensitive than the monitor-derived respiratory rate [18]. The algorithm uses an adaptive amplitude threshold to identify the timing of individual breaths within the IP signal, as shown in Figs 2 and 3. Prior to breath detection the IP signal is filtered and cardiac interference removed using the ECG signal [18,19]; this filtered IP signal is shown in Figs 2 and 3. From the IBIs, the respiratory rate was calculated in 20 second windows with increments of 1 second. In each window the respiratory rate was calculated as 60 divided by the mean IBI. To account for prolonged pauses in breathing overlapping the edge of the window (and leading to spurious results), if the first or last breath were greater than 1.5 seconds (equivalent to 40 breaths per minute) from the edge of the window then an additional IBI equal to the duration of the breath from the edge of the window was included in the IBI sequence for that window.

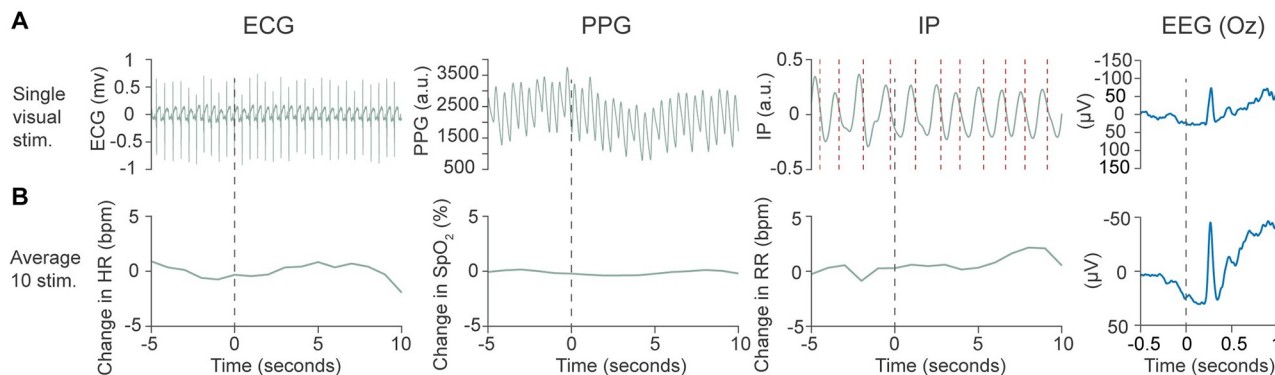

**Fig 2. Time-locked physiological responses to a visual stimulus in a single infant.** Electrophysiological and vital signs responses to a single visual flash stimulus (A) and averaged across a train of 10 visual stimuli (B). Change in vital signs are calculated by baseline correcting to the pre-stimulus mean. Blue signals are from the electrophysiological research recordings, green signals from the hospital vital signs monitor. Black dashed line indicates the point of stimulation. Red dashed lines on the IP indicate the time at which the amplitude threshold used for breath detection was crossed (see Methods—Analysis: Study 3 Demonstration of system utility). EEG responses are shown at the Oz recording electrode. Abbreviations: Stim.: Stimulus/stimuli, ECG: Electrocardiograph, HR: Heart rate in beats per minute, PPG: Photoplethysmography, a.u.: Arbitrary units, IP: Impedance pneumography, RR: Respiratory rate in breaths per minute, EEG: Electroencephalograph.

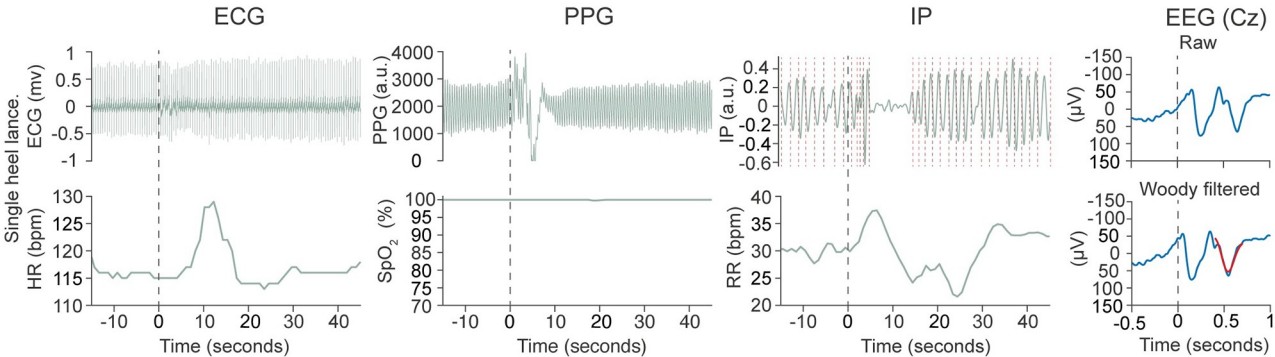

**Fig 3. Time-locked physiological responses to a clinically-required heel lance in a single infant.** Electrophysiological and vital signs responses to a single heel lance. Black dashed line indicates the point of stimulation. The EEG response is shown at the Cz recording electrode. The previously described template of noxious-evoked brain activity [16] is projected on to the EEG response and shown in red. Before the template is projected, the data is first Woody filtered (with a maximum of +/-100ms) to account for latency differences in the response. The raw trace (not Woody filtered) and Woody filtered traces are shown. Red dashed lines on the IP indicate the time at which the amplitude threshold used for breath detection was crossed (see Methods—Analysis: Study 3 Demonstration of system utility). Blue signals are from the electrophysiological research recordings, green signals from the hospital vital signs monitor. Abbreviations: ECG: Electrocardiograph, HR: Heart rate in beats per minute, PPG: Photoplethysmography, a.u.: Arbitrary units, IP: Impedance pneumography, RR: Respiratory rate in breaths per minute, EEG: Electroencephalograph.

## Results

### Study 1: Precision of time-locking of events to vital signs monitor

Events were time-locked to the recordings from the hospital vital signs monitor with a precision of 105 ms (range: 40–136, standard deviation: 19, N = 35 events, S2 Fig). The mean across the seven recording sessions ranged from 82 to 123 ms (N = 5 events per recording).

### Study 2: Precision of time-synchronisation between devices

The average (median) absolute time difference between event markers on the two recording devices was 115 ms (N = 51 recording sessions, interquartile range: 56 ms, range: 0–2993 ms—one recordings session was an extreme outlier, excluding this session the range of time differences was 0–334 ms). This is consistent with the results from Study 1; precision of time-locking to the electrophysiological recording equipment has previously been shown to be less than 1 ms [13]. As expected, due to differences in sampling rate and filtering between the devices, where multiple stimuli were recorded in a single session the time difference within the session between devices was low (average difference within a recording: 20 ms, range 0–68 ms, N = 15).

### Study 3: Demonstration of system utility

Simultaneous electrophysiological and vital signs recordings are presented in response to a visual stimulus in one infant (Fig 2) and a clinically-required heel lance in a separate infant (Fig 3), demonstrating the utility of the system in recording multimodal responses to a single stimulus and for the characterisation of average evoked responses.

## Discussion

Here we describe the development of a new device known as 'the PiNe box' and integrate it into a system to time-lock stimuli across multiple recording devices in infants, including the continuous data downloaded from the infant's hospital vital signs monitor. This enables

multimodal recordings, including electrophysiological and physiological activity, of responses to experimental and clinically-required stimuli in hospitalised infants at the cotside. By linking with the infant's hospital monitor, no additional probes or electrodes are necessary, thus reducing the time an infant is handled. This is particularly important in the youngest and sickest infants. Moreover, hospital staff are already familiar with existing equipment, thereby enabling easier integration of research into the clinical setting. We demonstrated the utility of the system by recording responses to visual stimuli and a clinically-required heel lance. However, the PiNe box can be used with a range of other stimuli including auditory and tactile stimuli [13], and to record other clinically-required procedures such as cannulation and immunisation [20–22]. To date we have successfully used the system in more than 50 research studies, including investigating the responses to experimental sensory stimuli and clinically-required heel lances. The PiNe box is not limited for use in infants but could be used in patients of any age, and is not restricted to the recording devices used here but could be used to link other network devices with recording equipment with TTL or similar communication ports (e.g. RS-232, USB) or extended to include data synchronisation from wireless devices.

We demonstrated excellent precision of time-locking of the stimuli to the recordings of the hospital vital signs monitor and synchronisation between recording devices. The event detection interface used here has been previously shown to have an accuracy of less than 1 ms to time-lock events to EEG recordings [13]. A precision of approximately 100 ms to time-lock stimuli to recordings of the hospital vital signs monitor provides sufficient accuracy for the analysis of changes in vital signs, which occur over a longer time scale than evoked EEG activity.

Multimodal recordings are essential to improve our understanding of the development of the nervous system in premature infants and could enhance clinical care, for example, through improving the assessment and treatment of pain [23–25]. Moreover, multimodal recordings of responses to noxious stimuli are essential to adequately assess the efficacy and safety of analgesics in randomised controlled trials [24] and easy-to-use acquisition methods which can work seamlessly alongside hospital monitoring devices will facilitate this [26]. The PiNe box is currently being used in the *Petal* randomised controlled trial assessing the benefits of parental touch for relieving pain in newborn infants [27], which provides a direct example of how this system can be used to easily acquire multimodal recordings in a clinical trial. In response to a clinically-required heel lance, the single infant shown in this paper has a clear physiological and neurophysiological response, including an increase in heart rate and an evoked potential in the EEG recording characterised well by the previously described and validated template of noxious-evoked brain activity [16]. The infant also has a short pause in breathing following the heel lance; short pauses in breathing occurred sporadically throughout the recording in this infant and investigation of whether this change in breathing is more likely to occur following a heel lance or occurred by chance here is beyond the scope of this paper. Infants can be more physiologically unstable following painful procedures [18,28], however, relatively little investigation of breathing patterns immediately after heel lances have been undertaken, perhaps in part due to inherent problems with hospital monitor-derived measures of respiratory rate in infants [18,19,29]. Whether changes in breathing patterns occur following sensory stimuli in infants warrants further investigation and could be achieved by combining the acquisition methods presented here with new analysis methods validated for use in infants [18,19].

## Conclusions

In summary, we describe an inexpensive and easily adaptable approach to time-lock stimuli across multiple recording devices by developing a new device based on a Raspberry Pi that

links network equipment, in particular a patient's vital signs monitor, with other network devices or those with ports such as TTL. This approach is relatively straightforward to implement and is cost effective. It is not limited to the stimuli presented here but could be used to time-lock a variety of both experimental and clinically-required stimuli. Characterising infant responses to stimuli at all levels of the nervous system is important to better understand development and improve clinical care.

## Supporting information

**S1 Fig. PiNe box pseudo code.** Running the Graphical User Interface, establishing connections, reading of the input stimulus triggers, generation of the UDP (User Datagram Protocol) output messages and transmission to the monitoring laptop, and switching on an LED on the PiNe box following stimulus triggers are all controlled by a bespoke module that is written fully in Python 3. The overall process of the code is summarised in this pseudo code.
(PNG)

**S2 Fig. Precision of time-locking of events to vital signs monitor.** To assess the precision of time-locking of events to the vital signs monitor (Study 2) the push button was pressed onto one of the ECG electrodes during recordings of background noise, generating a visible artefact on the IP (impedance pneumography) signal. The recording during 35 button presses is shown here, with each row a single recording session. The blue lines are the IP signal, time = 0 (black dashed lines) is the time of the event mark annotation from the push button (marked on the recordings via the PiNe box), red dot indicates the start of the artefact generated by the push button calculated as the point at which the signal went below a threshold defined as 3% of the minimum signal value in the 1 second after the event mark.
(PNG)

**S3 Fig. Precision of time-locking of events to vital signs monitor—ECG traces.** To assess the precision of time-locking of events to the vital signs monitor (Study 2) the push button was pressed onto one of the ECG electrodes during recordings of background noise. Whilst this always generated a visible artefact on the IP (impedance pneumography) signal (S2 Fig), there was only a large visible artefact on the ECG trace on some occasions. This difference is due to the inbuilt filtering of the Philips monitor which is different for the different signals. The recording during 35 button presses is shown here, with each row a single recording session. The blue lines are the ECG signal, time = 0 is the time of the event mark annotation from the push button (marked on the recordings via the PiNe box).
(PNG)

**S1 Table. Infant demographics for Study 2.** Values are median (lower quartile, upper quartile) or number (%). PMA = postmenstrual age. Apgar scores were missing from 2 infants.
(DOCX)

**S1 Data. Data file for Study 1.** Data files for Study 1, which consisted of 7 recordings where the push button device was manually pressed onto the ECG electrodes to investigate the time difference between the event annotation and recording artefact. Code to analyse this data is provided in S1 Code.
(ZIP)

**S1 Code. Code for analysis of Study 1.** MATLAB code for analysis of Study 1. The associated data are provided in S1 Data.
(M)

## Acknowledgments

We would like to thank Daniel Crankshaw, Annalisa Hauck, Fiona Moultrie, Shellie Robinson and Fatima Usman for help with data collection.

## Author Contributions

**Conceptualization:** Alan Worley, Kirubin Pillay, Maria M. Cobo, Caroline Hartley.

**Data curation:** Kirubin Pillay, Maria M. Cobo, Gabriela Schmidt Mellado, Marianne van der Vaart, Caroline Hartley.

**Formal analysis:** Alan Worley, Kirubin Pillay, Maria M. Cobo, Marianne van der Vaart, Caroline Hartley.

**Funding acquisition:** Caroline Hartley.

**Investigation:** Alan Worley, Kirubin Pillay, Maria M. Cobo, Gabriela Schmidt Mellado, Marianne van der Vaart, Caroline Hartley.

**Methodology:** Alan Worley, Kirubin Pillay, Maria M. Cobo, Gabriela Schmidt Mellado, Caroline Hartley.

**Project administration:** Maria M. Cobo, Gabriela Schmidt Mellado, Aomesh Bhatt, Caroline Hartley.

**Resources:** Alan Worley, Kirubin Pillay, Maria M. Cobo, Caroline Hartley.

**Software:** Alan Worley, Kirubin Pillay, Maria M. Cobo.

**Supervision:** Aomesh Bhatt, Caroline Hartley.

**Validation:** Alan Worley, Kirubin Pillay, Maria M. Cobo, Caroline Hartley.

**Visualization:** Alan Worley, Maria M. Cobo, Caroline Hartley.

**Writing – original draft:** Alan Worley, Caroline Hartley.

**Writing – review & editing:** Kirubin Pillay, Maria M. Cobo, Gabriela Schmidt Mellado, Marianne van der Vaart, Aomesh Bhatt, Caroline Hartley.

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
