## [Decision Letter · Decision Letter 0]

17 Apr 2023

PONE-D-23-03921The PiNe box: Development and validation of an electronic device to time-lock multimodal responses to sensory stimuli in hospitalised infantsPLOS ONE

Dear Dr. Hartley,

Thank you for submitting your manuscript to PLOS ONE. After careful consideration, we feel that it has merit but does not fully meet PLOS ONE’s publication criteria as it currently stands. Therefore, we invite you to submit a revised version of the manuscript that addresses the points raised during the review process.

We look forward to receiving your revised manuscript.

Kind regards,

Talib Al-Ameri, Ph.D

Academic Editor

PLOS ONE

Journal Requirements:

2. Please note that PLOS ONE has specific guidelines on code sharing for submissions in which author-generated code underpins the findings in the manuscript. In these cases, all author-generated code must be made available without restrictions upon publication of the work.

Please review our guidelines at https://journals.plos.org/plosone/s/materials-and-software-sharing#loc-sharing-code and ensure that your code is shared in a way that follows best practice and facilitates reproducibility and reuse.

"This work was funded by the Wellcome Trust and Royal Society through a Sir Henry Dale Fellowship (Grant reference: 213486/Z/18/Z).." 

"This work was funded by the Wellcome Trust and Royal Society through a Sir Henry Dale Fellowship awarded to CH (Grant reference: 213486/Z/18/Z). The funders had no role in study design, data collection and analysis, decision to publish, or preparation of the manuscript."

5. We note that you have a patent relating to material pertinent to this article:

"KP, MMC, AB and CH are inventors/contributors to Patent Application Number: PCT/GB2022/052836: SYSTEMS AND METHODS FOR MEASURING A RESPONSE OF A SUBJECT TO AN EVENT. The authors declare no other competing interests."

Please provide an amended statement of Competing Interests to declare this patent (with details including name and number), along with any other relevant declarations relating to employment, consultancy, patents, products in development or modified products etc. 

Please confirm that this does not alter your adherence to all PLOS ONE policies on sharing data and materials, as detailed online in our guide for authors http://journals.plos.org/plosone/s/competing-interests by including the following statement: "This does not alter our adherence to  PLOS ONE policies on sharing data and materials."

If there are restrictions on sharing of data and/or materials, please state these. Please note that we cannot proceed with consideration of your article until this information has been declared.

6. We note that you have indicated that data from this study are available upon request. PLOS only allows data to be available upon request if there are legal or ethical restrictions on sharing data publicly. For more information on unacceptable data access restrictions, please see http://journals.plos.org/plosone/s/data-availability#loc-unacceptable-data-access-restrictions. 

Reviewers' comments:

Reviewer's Responses to Questions

**Comments to the Author**

1. Is the manuscript technically sound, and do the data support the conclusions?

Reviewer #1: Yes

2. Has the statistical analysis been performed appropriately and rigorously? 

Reviewer #1: Yes

3. Have the authors made all data underlying the findings in their manuscript fully available?

Reviewer #1: No

4. Is the manuscript presented in an intelligible fashion and written in standard English?

Reviewer #1: Yes

5. Review Comments to the Author

Reviewer #1: This is a well written article that describes the validation of a new system to synchronise electrophysiological recordings with clinical vital signs monitors and event-mark both recordings simultaneously. This is a great system which would much improve research, and more importantly, reduce the time and handling required to set-up a study involving vulnerable neonates.

Major points:

Lines 183-189: Please clarify if the raw traces of the ECG, respiration and saturation also exported to the laptop, or just the pre-calculated rates. It isn't clear if these sampling rates pertain to the information received by the Philips monitor or the laptop, as it only explicitly states that the pre-calculated rates were downloaded.

If not, this is a limitation of the system, researchers may be interested in exploring heart rate in more depth, like HRV or presence of arrhythmias., and respiration traces are an important component of accurate sleep staging due to potential irregularities like apnoeas.

Lines 237-238: ‘The artefact was most visible on the impedance pneumograph and so this signal was used in the analysis.’

There is no mention of the pneumograph before this point. Not sure why there is an artefact on the pneumograph based on the description of the procedure. This must be described in the methods.

Lines 265-267: It is important to include more detail regarding the technical difficulties to clarify that the issue is not likely to occur again. Otherwise it is misleading to exclude an extreme value from this analysis. Also, without knowing what the difficultly was, this could be an issue that others may face, which is important information for a product that would be sold.

Minor points:

Abstract: the use of the term ‘noxious’ in ‘experimental visual, auditory, tactile and noxious stimuli’. I believe that the authors are referring to the 128mN pinprick stimulator but I think we should be careful calling this noxious and instead refer to this as a mechanical punctate stimulus. It is done experimentally so is perhaps not ethical to assume it is noxious. The conclusion that this stimulus is a less intense noxious stimulus compared to a skin breaking procedure, is built upon the use of template analysis using woody-filtering techniques that may force similarities between two different cortical responses, indeed there is no behavioural distress following the pinprick and the afferent fibre activation following the two stimuli will certainly be different. Moreover, the template assumes that this one event at a single electrode reflects the complexity of pain.

Lines 50-51: ‘a complete picture’, ‘better understand’. This is vague statement, think about expanding.

Lines 101-103: ‘the General-Purpose Input Output (GPIO) ports of the electrophysiological recording equipment (which allows simultaneous time-locking of stimulus onset to the electrophysiological activity).’

To help with clarity, it would be useful to add here that this is the connection which already exist.

Lines 191-199: You should state here how often you repeated the process. You provide this information for the other stimuli but only mention the number of repetitions for this study in the analysis.

Lines 251-256: As there is no mention of the acquisition filters applied by the Philips monitor, it is not clear if adding the extra 50Hz notch filter is sufficient to match the filters applied to both datasets.

Lines 277-278: This is very minor, but it would be nice to see the raw activity rather than the altered version, especially since the analysis of the brain activity is not relevant to the point being made by study 3.

6. PLOS authors have the option to publish the peer review history of their article (what does this mean?). If published, this will include your full peer review and any attached files.

Reviewer #1: No

---

## [Author Response · Author response to Decision Letter 0]

9 Jun 2023

Reviewer #1: This is a well written article that describes the validation of a new system to synchronise electrophysiological recordings with clinical vital signs monitors and event-mark both recordings simultaneously. This is a great system which would much improve research, and more importantly, reduce the time and handling required to set-up a study involving vulnerable neonates.

Thank you for your review. We have addressed them in the manuscript as outlined below, which has allowed us to substantially improve the paper.

Major points:

Lines 183-189: Please clarify if the raw traces of the ECG, respiration and saturation also exported to the laptop, or just the pre-calculated rates. It isn't clear if these sampling rates pertain to the information received by the Philips monitor or the laptop, as it only explicitly states that the pre-calculated rates were downloaded.

If not, this is a limitation of the system, researchers may be interested in exploring heart rate in more depth, like HRV or presence of arrhythmias., and respiration traces are an important component of accurate sleep staging due to potential irregularities like apnoeas.

Thank you for highlighting this important point. All raw traces were also downloaded. We have clarified this in the text which now states (lines 187-196):

Heart rate, oxygen saturation and respiratory rate (calculated by the monitor) were downloaded onto the laptop at a sampling rate of 1 Hz; the ECG was downloaded onto the laptop at a sampling rate of 250 Hz and recorded with three electrodes placed on the infant’s chest, the impedance pneumograph was downloaded onto the laptop at a sampling rate of 62.5 Hz and recorded from the chest electrodes, and the photoplethysmography was downloaded onto the laptop at a sampling rate of 125 Hz from a probe placed on the infant’s foot. The sampling rates of the signals are set by the Philips monitor rather than the software on the laptop; the PiNe box does not rely on certain sampling rates and could be used with clinical monitors with different sampling rates. Alternative software is available to download data from Philips or other monitors (14).

Lines 237-238: ‘The artefact was most visible on the impedance pneumograph and so this signal was used in the analysis.’

There is no mention of the pneumograph before this point. Not sure why there is an artefact on the pneumograph based on the description of the procedure. This must be described in the methods.

The impedance pneumograph signal is measured by the Philips monitor using the same electrodes for the ECG trace. When we pressed the push button onto the ECG electrode this generated an artefact on the ECG signal and the impedance pneumograph. The artefact was larger on the IP signal. We have clarified this in the text on lines 248-251 by stating:

As the push button was pressed on the ECG electrodes this generated an artefact on the ECG trace and the impedance pneumograph (which is a recording of changes in the resistance between ECG electrodes). The artefact was most visible on the impedance pneumograph and so this signal was used in the analysis (S2,3 Figs).

We have also added Supplementary Figure 3 which shows the ECG traces.

Lines 265-267: It is important to include more detail regarding the technical difficulties to clarify that the issue is not likely to occur again. Otherwise it is misleading to exclude an extreme value from this analysis. Also, without knowing what the difficultly was, this could be an issue that others may face, which is important information for a product that would be sold.

Thank you for your important point. We have now included this value in the analysis and revised the results accordingly (lines 331-334):

The average (median) absolute time difference between event markers on the two recording devices was 115 ms (N=51 recording sessions, interquartile range: 56 ms, range: 0 – 2993 ms – one recordings session was an extreme outlier, excluding this session the range of time differences was 0-334 ms).

Minor points:

Abstract: the use of the term ‘noxious’ in ‘experimental visual, auditory, tactile and noxious stimuli’. I believe that the authors are referring to the 128mN pinprick stimulator but I think we should be careful calling this noxious and instead refer to this as a mechanical punctate stimulus. It is done experimentally so is perhaps not ethical to assume it is noxious. The conclusion that this stimulus is a less intense noxious stimulus compared to a skin breaking procedure, is built upon the use of template analysis using woody-filtering techniques that may force similarities between two different cortical responses, indeed there is no behavioural distress following the pinprick and the afferent fibre activation following the two stimuli will certainly be different. Moreover, the template assumes that this one event at a single electrode reflects the complexity of pain.

You are correct that we were referring to the pinprick stimulator here. This stimulator is produced by MRC systems (Germany), and we refer to it as a noxious stimulator as the company produced the stimulator to activate nociceptors. However, to avoid confusion, as this stimulator is not used and not relevant for the current paper, we have removed reference to experimental noxious stimuli in the abstract.

Lines 50-51: ‘a complete picture’, ‘better understand’. This is vague statement, think about expanding.

This sentence has been revised to read (lines 50-51):

Investigating how an infant responds to a stimulus, at all levels of the nervous system, is essential to understand nervous system development and long-term effects.

Lines 101-103: ‘the General-Purpose Input Output (GPIO) ports of the electrophysiological recording equipment (which allows simultaneous time-locking of stimulus onset to the electrophysiological activity).’

To help with clarity, it would be useful to add here that this is the connection which already exist.

This has been added (line 110).

Lines 191-199: You should state here how often you repeated the process. You provide this information for the other stimuli but only mention the number of repetitions for this study in the analysis.

Thank you, we have now added this to this part of the methods as well (lines 208-209).

Lines 251-256: As there is no mention of the acquisition filters applied by the Philips monitor, it is not clear if adding the extra 50Hz notch filter is sufficient to match the filters applied to both datasets.

Philips do not describe the filters they apply during the acquisition. They have different modes and we used the default mode which is described as ‘monitor’ mode by Philips. We have added this to the manuscript, and clarified that we applied the 12-40 Hz filter after acquisition in line with the filter we also apply to the signal acquired through the Curry system. We have edited lines 264-269 to read:

The ECG signal from the Philips monitor (the clinical vital signs monitor) is filtered during acquisition using the settings built into the monitor itself (the filter was set to ‘monitor’ mode). To eliminate slow frequency drift due to movement artefacts and to make the filters comparable to those used for the ECG signal acquired using Curry 7 we additionally filtered the signal after acquisition from 12-40 Hz using EEGlab (15). The ECG signal acquired using Curry 7 (Compumedics Neuroscan - the electrophysiological research recording device) was filtered after acquisition from 12-40 Hz using EEGlab…

Lines 277-278: This is very minor, but it would be nice to see the raw activity rather than the altered version, especially since the analysis of the brain activity is not relevant to the point being made by study 3.

This has been added to Figure 3 – shown below.

---

## [Editor Report · Decision Letter 1]

29 Jun 2023

The PiNe box: Development and validation of an electronic device to time-lock multimodal responses to sensory stimuli in hospitalised infants

PONE-D-23-03921R1

Dear Dr. Hartley,

We’re pleased to inform you that your manuscript has been judged scientifically suitable for publication and will be formally accepted for publication once it meets all outstanding technical requirements.

Kind regards,

Talib Al-Ameri, Ph.D

Academic Editor

PLOS ONE

---

## [Editor Report · Acceptance letter]

6 Jul 2023

PONE-D-23-03921R1 

The PiNe box: Development and validation of an electronic device to time-lock multimodal responses to sensory stimuli in hospitalised infants 

Dear Dr. Hartley:

I'm pleased to inform you that your manuscript has been deemed suitable for publication in PLOS ONE. Congratulations! Your manuscript is now with our production department. 

Kind regards, 

on behalf of

Dr. Talib Al-Ameri 

Academic Editor

PLOS ONE